

**Long Range Prediction and the Stratosphere**
Adam A. Scaife[1,2], Mark P. Baldwin[3], Amy H. Butler[4], Andrew J. Charlton-Perez[5], Daniela
I.V. Domeisen[6,7], Chaim I. Garfinkel[8], Steven C. Hardiman[1], Peter Haynes[9], Alexey Yu
Karpechko[10], Eun-Pa Lim[11], Shunsuke Noguchi[12,13], Judith Perlwitz[14], Lorenzo Polvani[15,16],
Jadwiga H. Richter[17], John Scinocca[18], Michael Sigmond[18], Theodore G. Shepherd[5], Seok-
Woo Son[19], and David W.J. Thompson[20].
[1] Met Office Hadley Centre for Climate Prediction and Research, Exeter, U.K.
[2] College of Engineering, Mathematics and Physical Sciences, Exeter University, Exeter, U.K.
[3] Department of Mathematics and Global Systems Institute, University of Exeter, Exeter, U.K.
[4] NOAA Chemical Sciences Laboratory (CSL), Boulder, CO, USA.
[5] Department of Meteorology, University of Reading, Reading, U.K.
[6] ETH Zurich, Institute for Atmospheric and Climate Science, Zurich, Switzerland.
[7] University of Lausanne, Institute of Earth Surface Dynamics, Lausanne, Switzerland
[8] Fredy and Nadine Herrmann Institute of Earth Sciences, Hebrew University of Jerusalem,
Jerusalem, Israel
[9] Department of Applied Mathematics and Theoretical Physics, University of Cambridge, Cambridge,
UK
[10] Finnish Meteorological Institute, Helsinki, Finland
[11] Bureau of Meteorology, Melbourne, Australia
[12] Research Center for Environmental Modeling and Application, Japan Agency for Marine-Earth
Science and Technology (JAMSTEC), Yokohama, Japan
[13] Meteorological Research Institute, Japan Meteorological Agency, Tsukuba, Japan
[14] NOAA Physical Sciences Laboratory (PSL), Boulder, CO, USA
[15] Columbia University, Department of Applied Physics and Applied Mathematics, U.S.A.
[16] Department of Earth and Environmental Sciences, New York City, NY, U.S.A.
[17] Climate and Global Dynamics Laboratory, National Center for Atmospheric Research, Boulder,
Colorado, U.S.A.
[18] Canadian Centre for Climate Modelling and Analysis, Environment and Climate Change Canada,
Victoria, BC, Canada
[19] School of Earth and Environmental Sciences, Seoul National University, Seoul, Republic of Korea
[20] Department of Atmospheric Science, Colorado State University, Fort Collins, Colorado, U.S.A.

*Correspondence to:*     Adam A. Scaife (adam.scaife@metoffice.gov.uk)



**Abstract.** Over recent years there have been parallel advances in the development of stratosphere resolving numerical models, our understanding of stratosphere-troposphere interaction and the extension of long-range forecasts to explicitly include the stratosphere. These advances are now allowing new and improved capability in long range prediction. We present an overview of this development and show how the inclusion of the stratosphere in forecast systems aids monthly, seasonal and decadal climate predictions. We end with an outlook towards the future of climate forecasts and identify areas for improvement that could further benefit these rapidly evolving predictions.

## 1 Introduction

The climate system contains significant unpredictable variance and - for daily weather fluctuations at least - it is thought to have a deterministic predictability horizon of around two weeks due to the sensitivity of the evolution of the atmospheric state to small errors in initial conditions (Lorenz 1969) - the so-called 'butterfly effect'. Recent estimates (Leung et al., 2020; Domeisen et al., 2018) as well as tests of the predictability of midlatitude *daily* weather using the latest global prediction models (Zhang et al., 2019; Son et al., 2020) produce similar estimates for this predictability limit. However, this does not preclude skilful forecasts of the *statistics* (most notably the average) of conditions at long range beyond this timescale (e.g. Shukla 1981). This predictability owes its existence to slowly varying predictable components of the climate system in the ocean, and in some cases the atmosphere, as well as externally forced changes such as volcanic or solar variability effects (e.g. Kushnir et al., 2019). Some of the more prominent examples of stratospheric variability such as sudden stratospheric warmings and their interaction with the troposphere (Baldwin et al., 2021) and the quasi-biennial oscillation and its associated teleconnections (Scaife et al., 2014a) have been shown to fall into this predictable category, thereby providing relatively slowly varying conditions to guide the turbulent troposphere and hence provide long range predictability of conditions beyond the two-week limit.

The extension of long-range prediction systems to explicitly include representation of the stratosphere follows many years of development of stratosphere resolving general circulation models (GCMs). By the late 20[th] century many leading centres for climate research had started to include the stratosphere in versions of their GCMs (Pawson et al., 2000; Gerber et al., 2012). Much of the early model development was motivated by the discovery of the ozone hole in the 1980s (Farman et al., 1985) and the need for simulations of ozone depletion and potential recovery of the ozone hole following the 1987 Montreal Protocol, which required atmospheric models that represented both the atmospheric dynamics and chemistry of stratospheric ozone depletion (Molina and Rowland 1974; Crutzen 1974). In most cases this was achieved by adding further quasi-horizontal layers to the domain of existing climate models to extend their representation of the atmosphere to the stratopause or beyond (e.g. Rind et al 1988; Beagley et al., 1997; Swinbank et al., 1998; Sassi et al., 2002), while also incorporating key radiative (e.g. Fels et al., 1985), chemical (e.g. Steil et al., 1998) and dynamical (e.g. Scaife et al., 2000) processes.

The early development of so called 'high top' climate models, which represent the whole depth of the stratosphere, in general preceded the discovery of the main body of evidence that the variability of the stratosphere is not only affected by, but also interacts with the lower atmosphere and surface climate. Pioneering early studies suggested that the stratosphere might have direct effects on the troposphere and surface climate (e.g. Labitzke 1965; Boville 1984; Kodera et al., 1990, 1995; Haynes et al., 1991; Perlwitz and Graf 1995). In subsequent years, as reliable observational records lengthened and large enough samples of stratospheric variability were amassed it was unequivocally demonstrated that stratospheric variability precedes important tropospheric changes in the extratropics (Baldwin and Dunkerton 1999, 2001). There was debate about causality and whether the stratosphere really does affect the atmosphere below (e.g. Plumb and Semeniuk 2003). However, experiments where the stratosphere is perturbed in numerical models show changes in surface climate and reproduce similar patterns of response at the surface to those found in real world observations (e.g. Polvani and Kushner



2002; Norton et al., 2003; Scaife et al., 2006; Joshi et al., 2006; Scaife and Knight 2008; Hitchcock and
Haynes 2016, White et al., 2020). These involve changes to planetary scale waves and also baroclinic
eddies in the troposphere that are consistent with changes in baroclinicity near the tropopause (Kushner
and Polvani 2004; Song and Robinson 2004; Wittman et al., 2004, 2007; Scaife et al., 2012; Domeisen
et al., 2013; Hitchcock and Simpson 2014; White et al., 2020). Importantly, as we discuss below, the
same mechanisms also appear to be at work across a broad range of timescales (Kidston et al., 2015).
In recent years, motivated by the evidence of surface effects of stratospheric variability in the mid-
latitudes, the high-top model configurations used for stratospheric research were incorporated into
leading long-range prediction systems. This was initially done in test experiments, some of which were
internationally coordinated (e.g. Butler et al., 2016; Tompkins et al., 2017). However, a growing number
of operational systems are now producing ensembles of predictions at lead times of months or years
with coupled ocean-atmosphere models that extend to the stratopause or beyond; for example at
Environment Canada (Merryfield et al., 2013), the Met Office in the UK (MacLachlan et al., 2014), the
German Weather Service DWD (Baehr et al., 2015), the Japan Meteorological Agency (Takaya et al.,
2017) and the European Centre for Medium Range Weather Forecasts (Johnson et al., 2019). In the
following sections we document the emerging impacts and benefits of this new capability for surface
climate predictions at monthly, seasonal, and annual to decadal lead times.

## 102   2 The stratosphere and monthly prediction

The best-established phenomenon that gives rise to predictability of surface climate from the
stratosphere are the tropospheric circulation changes that follow strong and weak conditions in the
stratospheric polar vortex (Baldwin and Dunkerton 1999, 2001). For example, weak vortex conditions
such as those found in a sudden stratospheric warming (SSW, Baldwin et al., 2021) are typically
followed by a weakening and southward shift of the tropospheric mid-latitude jet stream (see e.g.
Kidston et al., 2015 and references therein) and thus the negative polarity of the North Atlantic
Oscillation (NAO), Arctic Oscillation (AO) and Northern Annular Mode (NAM). These fluctuations
also show a tendency to vacillate between strong westerly and weak (SSW) states on subseasonal
timescales (Kuroda and Kodera 2001; Hardiman et al., 2020a). The changes in the troposphere persist
roughly as long as those in the lower stratosphere, and last for around two months (Baldwin and
Dunkerton 2001; Baldwin et al., 2003; Hitchcock et al., 2013; Son et al., 2020; Domeisen 2019). The
impacts on surface climate also affect the frequency of extremes of temperature and rainfall (Scaife et
al., 2008; King et al., 2019; Cai et al., 2016; Domeisen et al., 2020b).
Although *major* SSW events, involving a complete reversal of the zonal flow in the mid stratosphere,
are rare in the southern hemisphere (Wang et al., 2020; Jucker et al., 2021), variations of the Antarctic
polar vortex are likewise followed by similar signatures in the underlying tropospheric flow, in this case
via the Southern Annular Mode (SAM). Weakening of the vortex is typically followed by a negative
shift in the SAM and associated changes in rainfall and near surface temperature (Thompson et al.,
2005; Lim E. et al., 2018, 2019, 2021). These changes in Southern Hemisphere circulation typically
take longer to reach the surface than their Northern Hemisphere counterparts (Graverson and
Christiansen 2003), perhaps due to the stronger stratospheric polar vortex and weaker wave driving in
the southern hemisphere, but they are nonetheless better predicted by improving stratospheric resolution
of forecast models (Roff et al., 2011). The timescale of weeks for the predictability of sudden warmings
is limited by the predictability of weather patterns in the troposphere which might trigger SSW events
(e.g. Mukougawa et al., 2005; Taguchi 2016; Garfinkel and Schwarz 2017; Jucker and Reichler 2018;
Lee et al., 2020a). However, if we add this timescale to the timescale of a month or more for the
persistence of lower stratospheric anomalies and their surface effects (e.g. Baldwin et al., 2003; Butler
et al., 2019), we arrive at the conclusion that on these occasions at least, initial conditions in the



atmosphere can provide predictability well beyond the usual two-week horizon for daily weather in
either hemisphere.
Predictability of the atmosphere at monthly lead times is also known to originate in part from the
Madden Julian Oscillation (MJO) in the troposphere and its teleconnection to the extratropics (e.g.
Vitart 2017). The circulation pattern associated with the MJO resembles a poleward and eastward
propagating Rossby wave with centres of action over the Pacific and extending into the Atlantic sector
where it also maps strongly onto the North Atlantic Oscillation. The lead time of around 10 days for the
impact of a change in the MJO to appear in the extratropical flow (e.g. Cassou 2008; Lin et al., 2009)
is also consistent with the timescale for poleward propagation of Rossby waves (e.g. Scaife et al., 2017).
It turns out that this tropospheric MJO teleconnection on monthly timescales also interacts with the
stratosphere (Garfinkel and Schwartz 2017). The MJO teleconnection to the North Pacific affects the
region most strongly associated with tropospheric precursors to SSW events, and consistent with this,
SSWs in the observational record have tended to follow certain MJO phases. The subsequent weak
vortex anomaly then propagates down to the troposphere (Garfinkel et al 2012), where it may strengthen
and prolong any existing negative NAO signal that is directly linked to the MJO via the troposphere
(Schwartz and Garfinkel 2017, 2020; Barnes et al., 2019).
In addition to the interaction of the MJO with the extratropical stratosphere, a further, completely
different link between the stratosphere and the MJO has recently been uncovered which modulates MJO
amplitude and persistence in the troposphere via the phase of the Quasi-Biennial Oscillation (QBO) in
the tropical lower stratosphere (Liu et al., 2014; Yoo and Son 2016; Martin et al., 2021). In this case,
easterly phases of the QBO appear to energise the MJO compared to westerly QBO, likely due to
changes in temperature and hence static stability close to the tropopause (Hendon and Abhik 2018;
Martin et al., 2019) with a potential contribution of cloud-radiation feedbacks (Son et al., 2017, see
Martin et al., 2021 for a review). This modulation of the MJO is in turn important for predictability as
it gives rise to higher monthly prediction skill of the MJO and its surface teleconnections during the
easterly phase of the QBO (Marshall et al., 2017; Abhik and Hendon 2019; Lim Y. et al., 2019).
Other mechanisms have also been found that can provide potentially predictable signals on the monthly
timescale. The traditional view of stratosphere-troposphere interaction involves upward propagation of
planetary scale Rossby waves (Charney and Drazin 1961), but this linear theory applies equally well to
downward propagation. Harnik and Lindzen (2001) and Perlwitz and Harnik (2003) identified a
possible source of downward propagating planetary waves in the form of reflecting surfaces in the
winter stratosphere. Examples of specific reflection events, showing upwards and then downward
propagation have since been observed (e.g. Kodera et al., 2008; Harnik 2009; Kodera and Mukougawa
2017; Mukougawa et al., 2017; Matthias and Kretschmer 2020). These results suggest that the details
of the stratospheric circulation such as regions of negative vertical wind shear could be important for
the formation of reflecting conditions (Perlwitz and Shaw 2013) and may yet provide a further
mechanism by which the stratosphere can affect the troposphere (Domeisen et al., 2019; Butler et al.,
168 2019).

Following studies demonstrating enhanced tropospheric predictability after SSW events in individual
climate models (e.g. Kuroda 2008; Mukougawa et al 2009; Marshall and Scaife 2010; Sigmond et al
2013), subseasonal forecast systems which explicitly represent the stratosphere in the climate system
were developed and implemented at operational prediction centres worldwide. It is often difficult to
demonstrate significant increases in overall skill (e.g. Richter et al 2020a) but routinely produced
ensembles of subseasonal predictions show that both stratospheric variability and its subsequent
tropospheric signature are predictable at monthly lead times (Domeisen et al 2020a, 2020b). The
strongest surface impacts occur if the polar vortex in the lower stratosphere is in a weakened state at
the time of the SSW (Karpechko et al., 2017) and there appears to be a roughly linear relationship
between the strength of these lower stratospheric anomalies and the tropospheric response (e.g. Runde



et al., 2016; White et al., 2020 and see Baldwin et al., 2019 for a review). We should note however that
there is no one-to-one correspondence between stratospheric variability and tropospheric events, and
some prominent examples of sudden stratospheric warmings are followed by differing tropospheric
anomalies (e.g. Charlton-Perez et al., 2018; Knight et al., 2020; Butler et al., 2020; Rao et al., 2020a).
Nevertheless, the canonical response is seen in the majority (~70%) of cases and periods of intense
wintertime stratospheric variability are important windows of opportunity to provide skilful monthly
forecasts (Mariotti et al., 2020; Tripathi et al., 2015a).

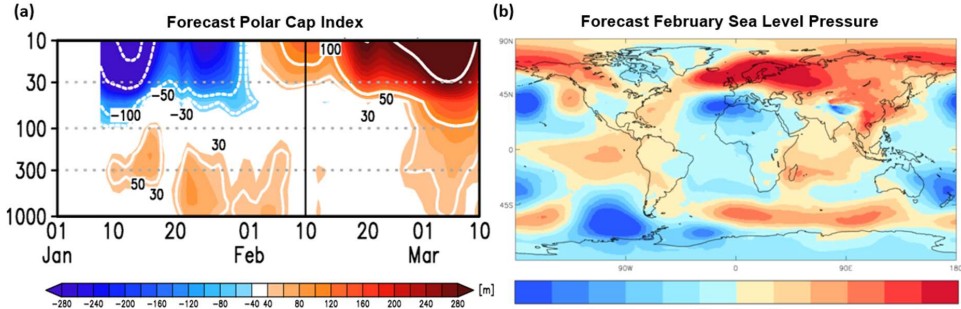

**Figure 1: Monthly forecasts prior to the 2018 sudden stratospheric warming and severe cold event over**
**northern Europe. Forecast Polar Cap Index (a) and February sea level pressure anomalies (b). Forecasts**
**were initialised in January 2018 (initialisation dates: 8th, 15th and 22nd) using the Met Office Hadley Centre**
**GloSea prediction system (MacLachlan et al., 2015). Sea level pressure is measured in hPa and Polar Cap**
**Index is the geopotential height anomaly (m) averaged over 65N to the North Pole.**
These forecast systems are now important tools for national meteorological and hydrological services
to monitor impending stratospheric variability and associated surface impacts in real time. Recent
extreme examples illustrate the importance of this activity. In February 2018 a major SSW occurred
and was followed by a strong negative NAO-like pattern at the surface with easterly wind anomalies
over Europe and multiple cold air outbreaks over the following weeks, including extreme snowfall
across northern Europe (Figure 1, Karpechko et al., 2018; Knight et al., 2020; Rao et al., 2020a) and an
abrupt end to Iberian drought in Southern Europe (Ayarzagüeña et al., 2018). Studies of monthly
ensemble predictions of this event with operational stratosphere resolving systems showed that the
stratospheric event was predictable at least 2 weeks in advance (Figure 1) and that the ensemble
forecasts indicated increased likelihood of cold surface conditions for several weeks after the event
(Karpechko 2018; Butler et al., 2020; Statnaia et al., 2020; Rao et al., 2020a). Again, as in the analysis
of previous events, there was also a strong association with the MJO entering Phase 7 with increased
convection in the West Pacific (cf. Garfinkel and Schwartz 2017) in the 2018 event. Finally, we should
also note that cases of monthly forecasts where the stratosphere plays an important role are not restricted
to winters with sudden stratospheric warmings and periods when the stratospheric polar vortex is above
normal strength also provide opportunities for skilful monthly forecasts (Tripathi et al., 2015b; Scaife
et al., 2016). In this case an opposite but symmetric surface response results, with strong *positive* NAO.
A very recent example occurred in February 2020, when, following an extremely strong polar vortex
(Hardiman et al., 2020b; Lee et al., 2020b; Lawrence et al., 2020; Rao et al., 2021a), the tropospheric
jet in the Atlantic sector strengthened, and the associated increased storminess and rainfall in this case
resulted in UK monthly rainfall reaching a new record high (Davies et al., 2021).

**3 The stratosphere and seasonal prediction**



Prior to the advent of dynamical forecast systems which explicitly represent the stratosphere, seasonal
forecasts using empirical relationships and statistical methods were proposed. These relied on the prior
state of the polar vortex and other predictable factors such as the QBO that are known to have links to
surface climate (Thompson et al., 2002; Charlton et al., 2003; Christiansen et al., 2005; Boer and
Hamilton 2008). In some cases they indicated additional predictability that was absent in existing
operational forecast systems, providing further evidence of predictability involving the stratosphere and
further motivating the extension of dynamical forecast systems to properly represent the stratosphere.
Similar empirical forecast studies continue, and although they cannot provide evidence of predictability
that is as strong as from forecasts using GCMs based on fundamental physical principles, they continue
to be useful to indicate sources of predictability that need to be properly represented in comprehensive
forecast systems (e.g. Folland et al., 2012; Wang et al., 2017; Hall et al., 2017; Byrne and Shepherd
227 2018).

Following the introduction of dynamical seasonal forecast systems with a good representation of the
stratosphere, clear links between successful seasonal prediction of the North Atlantic Oscillation, the
closely related Arctic Oscillation and the state of the stratospheric polar vortex have been identified in
forecast output (e.g. Scaife et al., 2014b; Stockdale et al., 2015; Jia et al., 2017; Byrne et al., 2019).
Similar signals are also seen in the southern hemisphere in relation to predictability of the Southern
Annular Mode (Seviour et al., 2014; Lim et al., 2021). Statistically significant increases in overall skill
directly attributable to the inclusion of the stratosphere in prediction systems is sometimes difficult to
demonstrate (e.g. Butler et al., 2016), especially given that other factors such as horizontal resolution
and physical parametrizations are often simultaneously changed. Nevertheless, the body of evidence
now weighs heavily in favour of predictability of the NAO and SAM from the stratospheric polar vortex
and from analyses showing reduced surface prediction skill in the absence of stratospheric variability
(e.g. Hardiman et al 2011; Sigmond et al., 2013; Scaife et al., 2016).
A second clear example of seasonal predictability originating in the stratosphere is the Quasi-Biennial
Oscillation (QBO). The QBO has such inherently long timescales that it persists for several months in
seasonal forecasts from initial atmospheric conditions alone and its regularity means that it can be
predicted from simple composites of earlier cycles. Nevertheless, a growing number of numerical
models used in seasonal forecast systems can now simulate and predict the oscillation within climate
forecasts (Garfinkel et al., 2018; Richter et al., 2020b; Stockdale et al., 2021) with the aid of forcing
from parametrized non-orographic gravity waves. A surface impact of the QBO is also well established
and has stood the test of time since it was first identified in the 1970s (Ebdon 1975; Thompson et al.,
2002; Anstey and Shepherd 2014; Gray et al 2018). Yet again this response projects closely onto the
North Atlantic Oscillation (and hence the Arctic Oscillation/Northern Annular Mode) in the northern
hemisphere, and the Southern Annular Mode in the SH. The favoured mechanism involves refraction
of vertically propagating Rossby waves in the lower stratosphere (Holton and Tan 1980), although other
pathways may also be involved (e.g. Inoue et al., 2011; Yamazaki et al., 2020; Rao et al., 2021b, 2021).
The observed magnitude of the QBO teleconnection is also large enough to provide seasonal
predictability of surface climate (Boer and Hamilton 2008) but its modelled amplitude at the surface
appears to be under-represented in current operational prediction systems and models (Scaife et al.,
2014b; Garfinkel et al., 2018; O'Reilly et al., 2019; Rao et al., 2020b; Anstey et al 2021).
In addition to the stratosphere acting as a source of predictability, other mechanisms by which the
stratosphere plays a role in seasonal predictions involve a pathway for global scale teleconnections.
These often originate in the tropics where the longer timescales of coupled ocean-atmosphere variability
such as the El Niño Southern Oscillation (ENSO, L'Heureux et al 2020) provide a predictable source
of low frequency variability. Effects on the extratropics can occur by tropical excitation of anomalous
Rossby waves which propagate polewards but also upwards into the stratosphere, as in the case of
ENSO (Manzini et al., 2006; Domeisen et al., 2019), giving two pathways for extratropical influence
(Butler et al., 2014). These highly predictable tropical sources of climate variability alter the strength



and position of the stratospheric polar vortex in the extratropics as well as the frequency of SSWs
(Polvani et al., 2017) and these are followed by changes in the seasonal westerly jets in the troposphere
and surface climate via the North Atlantic Oscillation (Ineson and Scaife 2009; Cagnazzo and Manzini
2009) or the Southern Annular Mode (Byrne et al., 2019). As might be expected, both the QBO and
ENSO teleconnections are best represented in seasonal forecast systems which contain a well resolved
stratosphere (Butler et al., 2016). We note that new examples of the stratosphere acting as a conduit for
seasonal teleconnections are still being uncovered (Hurwitz et al., 2012, Woo et al., 2015). For example,
the Indian Ocean Dipole (IOD) received little attention in this context until the recent record event of
late 2019, when it appears to have driven an extreme winter strengthening of the northern hemisphere
stratospheric polar vortex. This strengthening took many weeks to decay, giving rise to extreme yet
highly predictable conditions in the stratosphere and around the Atlantic sector in late boreal winter
(Hardiman et al., 2020b; Lee et al., 2020b). The same event was also implicated in extreme changes in
the polar vortex and the near SSW in the southern hemisphere (Rao et al., 2020e); an event that itself
likely helped to drive the extreme summer conditions and wildfires over Australia that year (Lim et al.,
279  2021).

Apparent links between Arctic sea ice and seasonal winter climate in the mid latitudes have also been
suggested to be mediated by the stratosphere, with increased Rossby wave activity and a weakening of
the stratospheric polar vortex in response to reduced sea ice, especially in the Barents-Kara Sea (Jaiser
et al., 2013; Kim et al. 2014; King et al., 2016; Kretschmer et al., 2016). Some studies also reproduced
surface signals in response to sea ice anomalies in seasonal forecasts of particular years that are in
apparent agreement with observational estimates (e.g. Balmaseda et al., 2010; Orsolini et al., 2012).
However, recent updates to observational records show weakening of these apparent effects (Blackport
and Screen 2020) and significant non-stationarity (Kolstad and Screen 2019). Subsequent modelling
studies with larger samples of simulations have provided mixed results (Zhang et al 2018; Dai and Song
2020; Smith et al 2021) and some argued that the atmospheric response to sea ice is weak and that while
the sensitivity to Barents-Kara sea ice may be stronger, the stratospheric response in particular is highly
variable (McKenna et al 2017). While there may well be a longer-term effect via the stratosphere from
sea ice decline (Sun et al., 2015; Screen and Blackport 2019; Kretschmer et al., 2020), sensitivity of the
response to the background state complicates the issue (Labe et al., 2019; Smith et al 2017), as do
possible confounding influences from the tropics (Warner et al 2020) and to date there is no clear
consensus for strong enough year to year effects to provide significant seasonal predictability.
Other proposed teleconnections acting via the stratosphere have been found in observations but remain
to be confirmed with successful reproduction in physically based climate models. A prominent example
involves a proposed link between Eurasian snow amounts and the stratosphere, followed by a return
influence on the NAO and surface climate. In this case, enhanced snow cover or depth is associated
with high pressure over north Eurasia, an increase in the flux of Rossby wave activity into the
stratosphere and a subsequent weakening of the stratospheric polar vortex, followed by the expected
negative shift in the NAO and AO (Cohen and Entekhabi; 1999, Cohen and Jones 2011; Cohen et al.,
2014; Furtado et al., 2015). However, the strength of this link in climate models and seasonal predictions
is modest (Fletcher et al., 2009; Riddle et al., 2013; Tyrrell et al., 2018, 2019) and does not agree with
apparent links to the AO in observations (Kretschmer et al., 2016; Garfinkel et al., 2020) even when
model mean state biases are corrected (Tyrrell et al., 2020). It has also been suggested that
teleconnections to snow are non-stationary or non-causal and there is continued debate about its long-
term robustness (Peings et al., 2013; Henderson et al., 2018).
In summary, a number of mechanisms by which the stratosphere acts to provide seasonal predictability
either by acting directly as a source of predictable variability (e.g. the QBO, SSWs), or as a conduit for
teleconnections (e.g. ENSO, MJO, IOD) have now been established in observations and have been
confirmed using climate model simulations based on first principles. These operate in seasonal forecast
systems, albeit with remaining errors such as the weakness of the QBO connection to surface climate.


Meanwhile, other mechanisms involving the stratosphere (for example the response to snow cover
variations) have been proposed based on apparent observed relationships, but until we have agreement
between these observations and theory (model simulations), scientists remain sceptical of whether they
represent actual sources of seasonal predictability and these topics remain topics of current research.

**4 The stratosphere and annual to decadal prediction**
In recent years, initialised predictions on longer timescales were developed on the premise of multiyear
memory in the ocean (e.g. Smith et al 2007), and following the development pathway mapped out by
seasonal forecasts in the past, these are now being run operationally to produce real time multi-model
forecasts (Smith et al., 2013). Kushnir et al., (2019) mapped out this operational development of annual
to decadal predictions and highlighted a number of sources of predictability, some of which involve the
stratosphere (Figure 2), but not all of which are fully represented in climate prediction systems.

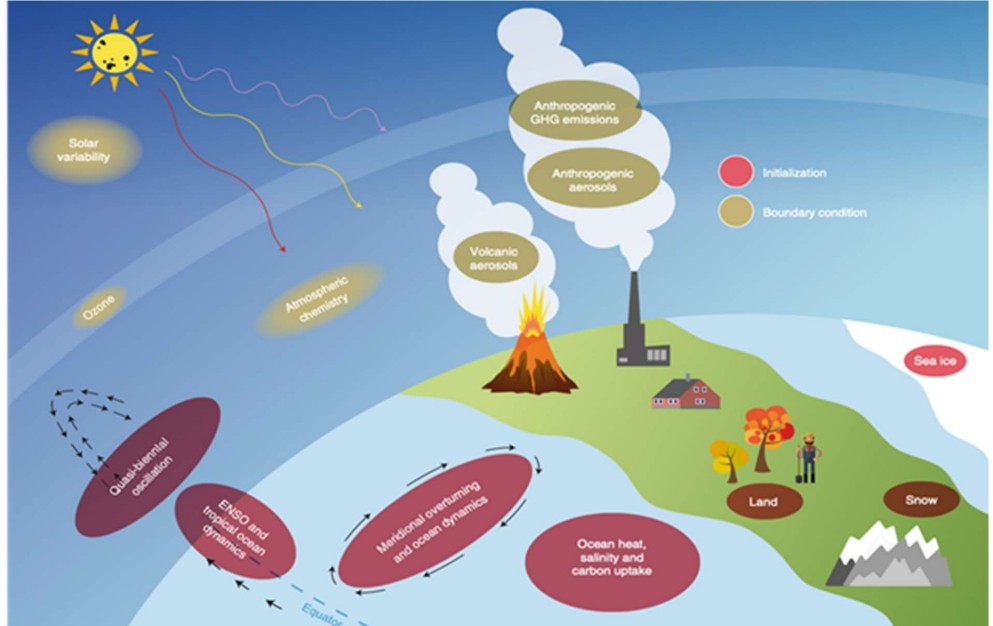


**Figure 2. Sources of annual to decadal predictability, some of which involve the stratosphere through the**
**response to external forcing, internal atmospheric dynamics, or ozone chemistry changes. After Kushnir et**
**al., 2019.**
Despite common misconceptions, not all annual to decadal predictability stems from the ocean. Indeed,
it has been clearly demonstrated that multiyear predictability of the QBO exists in current decadal
predictions systems out to lead times of several years (Pohlman et al., 2013; Scaife et al., 2014b). This
offers the prospect of a stratospheric contribution to multiyear predictability of the extratropics through
the teleconnection with the Arctic Oscillation (Anstey and Shepherd 2014, Gray et al., 2018) and to
tropical predictability through links to the MJO (e.g. Martin et al., 2021) and wider tropical climate
variability (Haynes et al., 2021).
Although it is more important on multidecadal timescales (see below), external forcing of the
stratosphere can also act as a source of decadal predictability. Forced climate signals from changes in
greenhouse gases or stratospheric effects such as ozone depletion occur on a much longer timescale



than the lead time of decadal forecasts but their contribution to the skill of predictions is not trivial. For
example, it is not immediately obvious whether the slow changes from multidecadal forced signals
would simply be swamped by unpredictable internal variability on decadal timescales, rendering long
term external forcing changes useless for decadal predictions. However, this is not the case and long-
term forcing is now known to be an important source of decadal prediction skill (Smith et al., 2019,
345 2020).

External forcing involving the stratosphere on shorter timescales is also important for annual to decadal
predictions. The stratosphere has long been known to be influenced by volcanic eruptions, particularly
in the case of tropical volcanic eruptions which are powerful enough to inject significant quantities of
sulphur dioxide into the atmosphere. Here it reacts with water to form sulphuric acid and persists in
aerosol form, leading to predictable multiyear global surface cooling, tropical stratospheric warming
and an intensification of the westerly stratospheric polar vortex in the extratropics (Robock and Mao
1992). Although the sample of observed events is limited, modelling studies have reproduced an
observed post-eruption intensification of the westerly winds in the stratosphere and some impacts on
the surface Arctic Oscillation. However, generations of models have struggled to reproduce the two-
year persistence of volcanic effects seen in observations and the observed magnitude of the effect on
the winter AO (e.g. Stenchikov et al., 2006; Marshall et al., 2009; Charlton-Perez et al., 2013, Bittner
et al., 2016). In addition to these changes in the atmosphere, the intensification of stratospheric
westerlies and hence Arctic Oscillation also combines with surface cooling of the ocean to generate
predictable changes in the Atlantic meridional overturning circulation (Reichler et al., 2012) which can
extend the volcanic influence to decadal timescales (Swingedouw et al., 2015). Finally, although the
mechanism is debated, there is also evidence of a multiyear effect of tropical volcanic eruptions on
ENSO, presumably requiring the persistent radiative forcing that arises through the long residence time
of volcanic products, particularly sulphate aerosols, in the stratosphere. This reportedly increases the
frequency of El Nino events by a factor of two in the years following volcanic eruptions (Adams et al.,
2003), again suggesting an important source of multiannual predictability via the stratosphere.
A second source of multiannual predictability from external forcing originates from solar variability
and in particular the 11-year solar activity cycle. Although a number of alternative mechanisms have
been proposed (see Gray et al., 2010 for a review), the established mechanism for surface effects via
the stratosphere is the change in the polar vortex that results from changes in upper stratospheric heating
over the course of each cycle between solar minimum and solar maximum. Atmospheric wave-mean
flow interactions amplify the initial radiatively driven change and drive its descent to the troposphere
(Kodera and Kuroda 2002; Marsh et al 2007; Ineson et al., 2011; Givon et al., 2021), where changes in
the extratropical jets result in a negative (positive) Arctic Oscillation pattern following solar minimum
(maximum). There is also evidence that it contributes to interannual prediction skill (Dunstone et al.,
2016) and an interesting aspect that has emerged in recent years is the integrating effect of the ocean on
solar induced changes in the NAO via interannual persistence of ocean heat content anomalies which
leads to a lag of around 3 years ($\pi/2$ cycles) in the peak response, as would be expected if the ocean is
integrating the effects of a periodic solar forcing (Scaife et al., 2013; Gray et al., 2013; Andrews et al.,
2015; Thiéblemont et al., 2015). However, debate continues as to whether the solar signal is indeed
large enough to be detectable in observations in the presence of large internal tropospheric variability
(Chiodo et al., 2019).
The currently recognised role of the stratosphere in decadal forecasts of surface climate again appears
mainly via the impact on annular modes and, in the northern hemisphere, the North Atlantic Oscillation.
Indeed, while much work is still needed to attribute variations in these modes to external forcing or
internal variations, current decadal prediction systems are now able to produce skilful predictions of
variations in the NAO on multiyear lead times (Smith et al., 2019, 2020; Athanassiadis et al., 2020).
These new results are important because they indicate new-found decadal predictability of events like
the high NAO of the 1990s which yielded a run of mild but wet and stormy winters in northern Europe





and the eastern USA. These winters are well known to have caused significant impact for example on
the insurance sector (Leckebusch et al., 2007) and coincided with the longest observed absence of SSW
events (Pawson and Naujokat 1999; Domeisen 2019). Given indications of coupled stratosphere-
troposphere variations on decadal timescales (Scaife et al., 2005; Omrani et al., 2014; Garfinkel et al.,
2017; Woo et al., 2015), understanding the role of the stratosphere in extratropical decadal predictions
needs further investigation.

**5 The stratosphere and multidecadal prediction**
The importance of the stratosphere for climate predictions on multidecadal timescales was generally
recognised before its role in predictions on shorter timescales. This is in part a legacy of the early
development of stratosphere-troposphere models for ozone depletion studies described in the
introduction and partly due to the later development of operational predictions for decadal timescales
for example.
Perhaps the best-known case for the stratosphere affecting multidecadal predictions of surface climate
is the influence of ozone depletion on the southern annular mode (SAM; Thompson and Solomon 2002,
2005; McLandress et al., 2011; Polvani et al., 2011; Son et al., 2018) where decreasing ozone in the late
20[th] century lead to a strengthened pole-to-equator temperature gradient, a stronger stratospheric polar
vortex and a shift to strong positive SAM phases at the surface. In this case, studies again show the
importance of stratospheric resolution to generate the full response, consistent with a genuine downward
influence (Karpechko et al., 2008). The associated poleward shift in the tropospheric jet is connected to
a delay in the spring breakdown of the stratospheric polar vortex (Byrne et al., 2017) and delivered
significant and prolonged changes in rainfall across many regions of the southern hemisphere (Kang et
al., 2011; Purich and Son 2012). Implementation of the Montreal Protocol in 1987 and subsequent
reductions in the rate of ozone depletion mean that recovery of the ozone layer is now expected over
the coming decades and the reversible effects of this on the surface climate form an important element
of current multidecadal predictions (Thompson et al., 2011; Previdi and Polvani 2014; Solomon et al.,
2016; Banarjee et al., 2020) where they are expected to play an important role alongside other changes
in the southern stratosphere due to continuing increases in greenhouse gases (Son et al., 2009; Barnes
et al., 2012; Ceppi and Shepherd, 2019).
The more limited effects of ozone depletion in the northern hemisphere meant that the role of the
stratosphere in multidecadal predictions took longer to become established. Some early studies found
potential amplification of positive Arctic Oscillation trends under climate change when the stratosphere
was included (Shindell et al., 2001). However, this was not borne out in later studies as simulations
with other fully coupled ocean-troposphere-stratosphere models suggested weakening of the
stratospheric polar vortex (e.g. Huebener et al., 2007). Subsequent studies with multiple models also
indicated a southward shift in the polar night jet with weakening high latitude winds and strengthening
subtropical winds (Scaife et al., 2012; Manzini et al., 2014). These changes result from increased
atmospheric wave driving of the winds which can overwhelm the cooling effect of greenhouse gases
(Karpechko and Manzini 2012) and can lead to important differences in future surface climate, for
example in regional rainfall in areas typically affected by the stratosphere via the Arctic Oscillation and
NAO (Scaife et al., 2012). There is still significant uncertainty due to the diversity of modelled
stratospheric responses to greenhouse gas increases (Manzini et al., 2014, Simpson et al., 2018, Zappa
and Shepherd 2017), and it has proved difficult to identify any clear change in the frequency of sudden
stratospheric warmings (Ayarzagüena et al., 2018, 2020; Rao et al., 2020c). This is perhaps due to the
competition between strengthening latitudinal temperature gradients near the tropopause and enhanced
meridional overturning in the mid stratosphere. There is also strong inherent unpredictable variability
from decade to decade in the frequency of SSW occurrence (Butchart et al., 2000; McLandress and
Shepherd 2009).



Other aspects of future climate change where the stratosphere plays a role have also been identified, for
example, in the debate over the response to future levels of Arctic sea ice. In this case it seems that the
response of the mid-latitude circulation involves a negative shift in the Arctic Oscillation (Screen et al.,
2018; Zappa et al., 2018; McKenna et al., 2018). This could again be amplified by interaction with the
stratosphere as some studies suggest that the stratospheric response is necessary for a large surface
response (Kim et al., 2014), while others highlight that the stratospheric interaction is sensitive to the
regional pattern of sea ice decline (McKenna et al., 2018), and still others show evidence of non-linear
stratospheric, and stratosphere-mediated surface response (Manzini et al., 2018), coincident with the
time when Barents and Kara seas become ice-free (Kretschmer et al., 2020). Furthermore, studies also
indicate that the surface climate response to sea ice decline depends systematically on the phase of the
stratospheric QBO (Labe et al., 2019).
Although it is much less certain than anthropogenic climate change, there have also been suggestions
of a multidecadal decline of external solar irradiance which can impact multidecadal climate predictions
via the stratosphere. Previous multidecadal solar minima, so called 'grand minima', have occurred in
sunspot records and have been connected to the Little Ice Age period around the end of the 17th century
using proxy and other data (Owens et al., 2017). Given recent weak amplitude 11 year solar cycles,
there are now suggestions of a future solar 'grand minimum' where the 11 year cycle described above
could become muted or even absent for a prolonged period (Lockwood et al., 2010). In this case, the
upper stratospheric cooling in the tropics and summer hemisphere can change the meridional
temperature gradient in a similar fashion to the 11 year cycle (Maycock et al., 2015) and leads to a
negative shift in the AO, the NAO, and hence affects regional climate (Ineson et al., 2015). However,
in this case it appears that while regional changes could be significant, they are generally much smaller
than the surface warming due to anticipated levels of anthropogenic greenhouse gases (Anet et al., 2013;
Ineson et al., 2015; Maycock et al., 2015).
Finally, we note that although frequency variability in teleconnections is observed (e.g. Garfinkel et al.,
2019) it is often unclear whether this is a systematic variation or simply due to sampling variability of
an underlying stationary process (Jain et al., 2018). There is also growing evidence for systematic
climate change in some of the teleconnections by which the stratosphere enables surface predictability.
Under future climate change it appears that some of the teleconnections discussed above may *strengthen*
in amplitude. For example, the connection between ENSO and the extratropical Atlantic/European
sector increases in future climate projections (Müller and Roeckner 2006; Fereday et al., 2020).
Similarly, recent analyses suggest that the MJO teleconnection to the extratropics increases under
climate change (Samarasinghe et al., 2020). The same is also true of the extratropical effects of the
stratospheric QBO, where in this case, the strength of the teleconnection doubles under future climate
change (Rao et al., 2020d) despite the QBO itself becoming weaker (Richter et al., 2020c).
Providing they are not swamped by changes in interannual variability, these strengthening
teleconnections suggest a growing importance of the stratosphere in surface climate prediction over the
coming decades.

**6 Outlook**
Long range prediction has evolved quickly in recent years (Merryfield et al., 2017; 2020, Butler et al.,
2019; Meehl et al., 2021) and this rapid development is due in part to the improved representation of
stratospheric processes and stratospheric initial conditions in ensemble prediction systems. The long-
range forecast community originally focused on predictability from initial ocean conditions and this
remains the primary source of long-range predictability, for example from ENSO, but some of these
long-range prediction systems contained poor representations of the stratosphere. In the meantime,
those working in parallel on climate modelling of the stratosphere were rarely involved in initialised



long-range prediction, instead being driven primarily by the ozone depletion problem. Knowledge
exchange across fields is important in science and precursors to a new paradigm often occur when a
topic is investigated from researchers from outside the field (Kuhn 1970). The crossover and
collaboration between long range prediction and stratospheric research communities is no exception
and has yielded rapid progress and new insights. Examples where initial atmospheric conditions can
provide predictability beyond the usually assumed limit have been demonstrated, particularly for the
extratropics but also for the tropics, and we now know that in some situations initial conditions in the
ocean have less impact than initial conditions in the stratosphere (Thompson et al 2002; Scaife and
Knight 2008; Polvani et al 2017). This suggests that initial atmospheric conditions are likely to be more
important for long range forecasts than previously assumed (Mukougawa et al., 2005, 2009; Stockdale
et al., 2015; Noguchi et al., 2016, 2020a; Choi and Son 2019; O'Reilly et al., 2019; Nie et al., 2019),
not least because the overturning and breaking of Rossby waves in the stratosphere is followed by long
lived atmospheric anomalies due to synoptic scale eddy feedbacks that prolong the effects in the
troposphere (Kunz and Greatbatch 2013) and enhance long range predictability (Kang et al., 2011).
A notable simplification to understanding the role of the stratosphere, at least in extratropical long-
range predictions, is its apparently seamless mechanism across different timescales and different
phenomena. Following the early ground-breaking studies showing surface impacts of stratospheric
variability and a multitude of studies on individual teleconnections between the stratosphere and surface
climate, the projection of stratospheric impacts onto the Arctic Oscillation/North Atlantic
Oscillation/Annular Mode circulation patterns across timescales and hemispheres is now well
established (see the review by Kidston et al., 2015). This suggests that similar coupling processes occur
between the stratosphere and troposphere from months to decades and these processes are responsible
for some of the most intense extratropical climate extremes, in winter in the NH and in late spring/early
summer in the southern hemisphere (Karpechko et al., 2018; Fereday et al., 2012, Kautz et al., 2019;
Domeisen and Butler 2020).
Some, but not all, leading forecast systems now include a well resolved stratosphere with reasonable
representation of relevant processes such as the body force from sub-grid orographic and non-
orographic gravity waves. However, many outstanding problems remain. Although their number is
increasing, only a subset of current GCMs have the ability to simulate a realistic QBO beyond its decay
from initial conditions and it seems that all GCMs have problems with the fidelity of modelled QBO
teleconnections, which are either too weak or absent altogether (Scaife et al., 2014a; Kim et al., 2020;
Anstey et al., 2021). Even the relatively well studied ENSO teleconnection via the stratosphere to the
extratropics still has outstanding questions, such as whether the stratosphere exhibits more SSW events
during the La Niña phase (Butler and Polvani 2011; Song and Son 2012). This is not generally
reproduced in modelling systems (Garfinkel et al., 2012) but occurred again in the most recent winter
at the time of writing in 2021. Similarly, while the increased monthly predictability from the MJO
during the easterly phase of the QBO has been detected in monthly forecast experiments, the QBO-
MJO connection does not persist in longer predictions and simulations with current models (Kim et al.,
2020). Research and model development on stratosphere-troposphere interaction, including tropical
effects (Noguchi et al., 2020b), will no doubt lead to further progress in resolving this issue (Haynes et
al., 2021).
In addition to teleconnection errors, mean biases in stratospheric climate are inevitably present to
varying degrees. The common protocol of running a set of retrospective predictions to allow these mean
biases to be estimated and hence subtracted from real time predictions may well correct for much of
this error. However, the degree to which biases have a nonlinear, state dependent impact on the
predictions is not fully understood. In some contexts, the nonlinear impacts of biases may be minimal
(Karpechko et al., 2021) while others show sensitivity (Sigmond et al., 2008, 2010) and increases of
prediction skill occur under certain background conditions, for example during Easterly QBO phases
(Taguchi 2018). Other processes generally omitted from long range predictions include interactive





variations of ozone and other trace gases. Although reports of impacts and benefits have varied, it is
thought that surface signals on interannual timescales come mainly from dynamical rather than
chemical changes (Seviour et al 2014; Harari et al., 2019). Nevertheless, some studies suggest
detectable effects from interannual variability of ozone and it may be that ozone fluctuations could help
to amplify surface signals (Karpechko et al., 2014; Son et al., 2013; Smith and Polvani 2014; Oehrlein
et al., 2020; Hendon et al., 2020), providing a further area for future development. Given that the cost
of full atmospheric chemistry schemes remains computationally expensive, it seems likely that simple
parametrizations of ozone chemistry (e.g. Monge-Sanz et al., 2021) would be valuable in this context.
We end with a pointer to an issue that has now been found to affect all long-range predictions from
monthly to seasonal to decadal and multidecadal timescales. So called 'perfect model studies', which
test the ability of models to predict their own ensemble members, are now known to *underestimate* the
true predictability of climate in some regions, particularly around the Atlantic basin and so models are
better at predicting real world variations than they are at predicting themselves – the so called 'Signal
to Noise Paradox' (Scaife and Smith 2018). This is surprising, because perfect model prediction scores
are often assumed to represent an upper (rather than lower) limit for prediction of the real world.
Whether the stratosphere is involved in the cause of this problem remains to be seen, as it appears first
in the troposphere (Domeisen et al., 2020a) and studies are undecided whether predictions of the
stratosphere exhibit the same issue (Saito et al., 2017; Stockdale et al., 2015). Nevertheless, the same
signal to noise issues may well be the reason for the weaker than observed amplitude of many modelled
tropospheric teleconnections involving the stratosphere. Resolving this problem will therefore likely
amplify these signals, provide greater levels of prediction skill, and further strengthen the role of the
stratosphere in long range predictions of surface climate.
**Author Contributions**
AAS wrote the draft manuscript. All other co-authors contributed relevant references and input to
revisions and editing of the manuscript. SWS provided Figure 1a.
**Acknowledgements**
AAS and SCH were supported by the Met Office Hadley Centre Climate Programme funded by BEIS
and Defra. MPB was supported by the Natural Environment Research Council (grant number
NE/M006123/1). JHR was supported by the Regional and Global Model Analysis (RGMA) component
of the Earth and Environmental System Modeling Program of the U.S. Department of Energy's Office
of Biological & Environmental Research (BER) via NSF Interagency Agreement 1844590. SN was
supported by the Japan Society for the Promotion of Science (KAKENHI, Grant Number: 19K14798).
EPL was supported by the Australian government's National Environmental Science Program Phase 2.
SWS was supported by the National Research Foundation of Korea (NRF) grant funded by the South
Korean government (Ministry of Science and ICT) (2017R1E1A1A01074889). DWJT is supported by
the US National Science Foundation Climate and Large-Scale Dynamics program. Support from the
Swiss National Science Foundation through project PP00P2_170523 to D.D. is gratefully
acknowledged. CIG was supported by an European Research Council starting Grant under the European
Unions Horizon 2020 research and innovation program (Grant agreement 677756).

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
