# Peer review of "Long Range Prediction and the Stratosphere"

_Atmospheric Chemistry and Physics, 2021_

## Author Response (AR1)

Responses to ACP review comments:

**CC1:** I would think that any long range prediction of the stratosphere would begin with the QBO, and even though the paper says "... its regularity means that it can be predicted from simple composites of earlier cycles", there remains no consensus on a first principles understanding of the underlying QBO synchroniation mechanism. After the QBO disturbance of 2016 died down, it appears that the regularity of the previous cycles returned, indicating that the synchronization is externally applied and not a natural resonance (in the latter case, a phase shift would occur). So much like a storm surge will only generate a transient in a tidal analysis, the QBO is also likely snchronized to an external tidal forcing, only transiently perturbed by ENSO disturbaces.

In the attached figure, the power spectrum of the QBO 30 hPa time series is shown. Spectral peaks are identified as below, aliased against a strong seasonal modulation.

1 = monthly Draconic tide

2 = monthly Tropical (18.6y modulation of Draconic)

3 = fortnightly Draconic (harmonic of #1)

4 = annual cycle

5 = semi-annual

6 = strong aliased harmonic of #1

**1 and #6 rising above the background is a strong substantiation. The basis of the underlying theory is described in the following cite: Pukite, P., Coyne, D., & Challou, D. (2019). Mathematical Geoenergy: Discovery, Depletion, and Renewal (Vol. 241). John Wiley & Sons.**

Thank you for this interesting comment. We agree that the QBO is a very important component of the stratosphere for long range prediction. We therefore devote substantial sections of the review to the QBO, including its inherent predictability and its surface impacts. Having said that, it does not appear first in the list of topics in our review because we take the approach of building up timescales from monthly to seasonal to multiannual and beyond and the QBO is most relevant to seasonal and interannual prediction.

Regarding your second point about the mechanisms and your statement that there "remains no consensus on a first principles understanding of the underlying QBO" and that the recovery of QBO oscillations following the 2016 disruption indicates that there is an external forcing mechanism:

Since the work of Lindzen and Holton (1968) an overwhelming mass of evidence has accrued in favour of the internal wave driven momentum flux mechanism for the QBO. Current global circulation models are now able to simulate the QBO oscillation period, amplitude, spatial structure and even fluctuations in the length of different cycles without resorting to external forcing. Similarly, long range forecasts initialised during the QBO disruption were also able to successfully predict the recovery and phase of the oscillation without the need for external forcing (Osprey et al., Science, 2016).

**RC1:**

A nicely written opinion piece that merits publication after some revision.

Thank you for this encouraging summary and for the following insightful comments.

After reading the paper, I was left slightly wondering: Is the paper trying to do too much or too little? It reads like a very nice essay, but somehow, I was wondering if the framing was too narrow or too wide. Reading the paper further, I found it hard to grasp how the authors suggest to handle the seamless nature of weather and climate modelling in the future (is there a recommendation?), and the transition from initial value dominated problems to boundary value dominated problems could and should be clearer. Of course, the main point of the paper is to discuss the usefulness of including the stratosphere into (atmospheric) predictions of weather and climate. However, is there still anybody left who doubts the usefulness of such an approach?

We agree that the transition from initial value to boundary value problems could be made clearer and so have altered the text at lines 56-66 and line 112-114 to emphasize this transition as the lead time of predictions increases. We have also added a new schematic (Figure 1) to more clearly illustrate whether the particular mechanisms discussed here occur mainly through initial conditions or boundary conditions.

Regarding the main point of the paper and whether anyone still doubts that it is necessary to include the stratosphere explicitly in prediction systems: a growing number of systems do now include the stratosphere but a good number do not. There is also still debate about how much this enhances the quality of forecasts, which is always relevant in any decision about where to most optimally spend computational resource (e.g. vertical extent vs horizontal resolution). We therefore think that now is a good time to document the evolution of the science to this point. Please also note that this paper was invited by the editors to contribute this topic to the associated 'Encyclopedia of Geosciences'.

Technically, when assimilating (satellite) data with wide vertical weighting functions the usefulness seems obvious (presumably this should be mentioned more strongly) – otherwise no good initial state for any kind of prediction could be generated.

Regarding the recommendation to mention deep weighting functions for data assimilation that extend into the stratosphere: we agree this is an important role for the stratosphere in atmospheric data assimilation and now mention it briefly at lines 102-106. However, as this review is about long-range prediction, we do not want to extend this point further, for example into the weather forecasting effects.

From fluid dynamical understanding, when e.g. thinking about wave propagation in the atmosphere, the case seems settled as well and many good and valid examples are given in the paper. In terms of coupling between composition, thermal structure and circulation, evidence exist on different scales as well, e.g. for volcanic eruptions (tropical presumably more than extra-tropical) or the role of the ozone hole for the seasonal evolution of surface temperature in Antarctica (presumably this could be addressed clearer in the paper).

We now emphasize the coupling between composition, thermal structure and circulation at lines 61 and lines 426-427.

In particular the role of the land surface (and its changes – including the hydrology) in providing – on some time horizon – an added benefit for prediction seems missing. How this is link to the stratosphere is presumably less clear-cut than the role of the ocean (I understand this), however I found this a strange omission that should be addressed (at the moment it is sort of mentioned as a caveat – maybe it could be mentioned more as a research need).

As this review is specifically about the role of the stratosphere in long range predictions rather than the land surface per se, and since we are not aware of a large body of evidence showing an influence of the stratosphere on long range predictions via the land surface (beyond the example of snow cover which is mentioned but still very much debated), we did not add further material on this topic.

The paper follows the philosophy by structuring the content by time horizon – however, given the review nature of the paper the time horizons should be clearly motivated in the introduction and a small sketch that illustrates the transition from the initial value to the boundary value regime should be included. I know that such figures exist. However, I believe it would be useful to start with a clear map to motivate the structure of the paper and to provide a caveat for decadal to multidecadal prediction.

As suggested, we have now added a schematic (Figure 1) showing the transition from initial conditions to boundary conditions in the stratosphere and how they may affect long range predictions. Thank you for this useful suggestion.

I have to admit that I personally have a problem with the use of the word "prediction" on such a long timescale, because the prediction will strongly depend on the chosen scenario. I would actually prefer the use of the term "projection" (over prediction) to make this more obvious to the uninitiated reader, that most of the "answer" (projected state at the end of the integration) might be actually in the scenario.

We have now altered 'prediction' to 'projection' on multidecadal timescales throughout the paper, starting for example with line 41 in the revised abstract.

The stratosphere and monthly prediction

I am surprised that so much space is given to the MJO – I would have assumed that the ENSO state would be even more fundamental (even on this timescale) – also regarding its connection to the MJO occurrence. Presumably ENSO – similar to the QBO – can be seen in some cases as something that provides a certain persistency to the system. If the initial state is correctly captured in the analysis that is used at the start of an integration the resulting prediction should benefit from the "accuracy" of the initial state. Here, I would have expected more emphasise on the initial state (ocean, land, QBO, BDC, …) and benefits that result from kicking-off the "forecast model" in the correct way.

We purposefully chose to emphasize the MJO and SSW events as these are the main two phenomena that involve the stratosphere and have been clearly identified as providing monthly predictability. The QBO and ENSO are given equal attention but as they have longer timescales they are mostly dealt with in the seasonal and interannual sections. Although we do not discuss the land surface or ocean per se because the review is specifically focused on the role of the stratosphere, we now note that there is relatively little information on the role of the accuracy of the initial stratospheric state on long range predictions and now mention this in the outlook section (Lines 523-524) as a topic that could benefit from more research.

The stratosphere and seasonal prediction

Of course, the monthly to seasonal scales are fluent (or seamless). Presumably this is now the range were initial conditions become less important and some boundary conditions count in more. Thus, I am surprised that the QBO is featured in this context stronger (my subjective feeling, line 241) as before. I would have assumed that transition timescales of the QBO are well within the scale range considered here, and that most models still perform poorly for the phase transition, even those models having some persistence when started with the right initial conditions.

We emphasize the QBO in this section as it has high predictability on this timescale and contributes strongly to seasonal and interannual predictions. Its phase transitions can also be predicted with reasonable skill. We now include this point at line 259 with associated references.

Given the ENSO link discussed here, I was wondering if the authors would like to comment on the question to what extent one can assume certain sources of predictability as independent. ENSO and MJO seem to be so closely linked (in some aspects) that they might more reflect the seamless nature of the problem than independent added benefits of predictability. To some extent lines 309-312 pick-up this point (stratosphere as source or conduit of predictability) – however at the same time I find the closure of the section confusing. What precisely are the first principles? Even the best models are built "only" on discretised versions of a certain set of coupled partial differential equations (close to first principles). However, the resulting model is an approximation that allows idealised model integrations that can be valid as sensitivity studies. Thus, often consistency and not causality can be tested – and I assume this is true for a forecast problem as well. Thus, I would phrase this sentence more carefully.

We agree that although forecasts often only provide consistency, GCM experiments with perturbations are a more powerful tool to establish causality and so we rephrased this at lines 235-236 to refer to 'GCM experiments' rather than forecasts. We do not elaborate further at this point to avoid disrupting the flow of the discussion, but examples were already given in the introduction at lines 93-94.

The stratosphere and annual to decadal prediction

Figure 2 seems to be more misleading than helpful – I appreciate that this is an adaption of an already existing figure. However, the classification of boundary conditions and initial state seems not very clear to me (in particular thinking through the role of the carbon cycle – including how composition responds to emission changes). Even though halogen loading might be considered as a boundary condition resulting from prior emissions, certainly knock on effects, including ozone, are not as clear cut (affected by other emissions and sometimes - also at lower chlorine loading than now - subject to sometimes "unusual" variability). In particular the climate (and carbon uptake in the SH) will change with ozone recovery, which is something that will happen in the near future on a decadal timescale. Thus, it would be nice to have a more critical reflection on the figure or an alternative approach (preferred).

We agree that the distinction between boundary and initial conditions can become blurred if they interact and the reviewer makes a fair point. However, some distinction is useful here as it is relevant to the forecast problem and setting up forecast systems which requires initial (~internal) conditions and boundary (~external) conditions to be specified. There are also fairly clear examples of primarily initial condition (e.g. QBO) and external (e.g. volcanic forcing) sources of predictability. Nevertheless, we acknowledge that these can interact and that this clouds the picture so this is now discussed at lines 411-413.

The stratosphere and multidecadal prediction

As already mentioned, I feel a little uneasy that in a projection context (scenario dependent) the word prediction is used. I understand that a prediction is certainly something that can depend on assumptions (e.g. scenarios). However, I have the feeling that many people judge uncertainties different, depending on a prediction or projection framing. That said, this could be rectified by a much clearer statement regarding the transition towards a boundary value problem that is dictated (to an important extent) by the chosen scenario in the introduction. In line 470 I find the "doubling strength

of a teleconnection" a weird concept, please explain precisely what is meant. In line 472-474 I am slightly lost: Why is there just talk of strengthening teleconnections – is it no longer not an open point if the explained variance (e.g. of a certain EOF) could change? For example, to stay in the EOF picture: Will the order of EOFs change?

We now use the word projection throughout as recommended by the reviewer.

The doubling strength simply refers to the fact that composites and similar measures show up to twice the amplitude in the future climate than the current climate irrespective of any change in the EOFs which we agree is plausible but is not discussed in our referenced sources. This is now explained at line 487-498.

Outlook

I am surprised by the interpretation summarised in lines 489 – 497. I am not doubting the general assumption – that the initial state of the stratosphere is important on certain timescales – however, also following the discussion above regarding ENSO and MJO more detail seems to be necessary, to explain the special situation and time horizon for which this statement is true.

It is true that it is only in certain examples that the effect of atmospheric intialisation is stronger than the ocean effect. This is particularly true of the monthly timescale when SSW events occur and this is now made clear at line 515-517.

I guess it would be charming to cite in line 500 really some ground-breaking early studies – and not just the meta-citation that follows.

Line 528 now cites two of the very early studies but please advise if there other relevant earlier papers that we have missed.

Lines 525-528 presumably requires a clear distinction between mean biases and teleconnection errors. The former seems to be used in the meaning of a classic bias (systematic deviation from a reference), the latter is presumably linked to a change in variance or order of the EOFs (or shape/phase/shift of the probability density functions, PDFs). Presumably both biases / changes are not independent – however, they are not the same either.

The definition of mean bias is now clarified at Lines 553-554.

Line 550-552 seem to be out of context – I am not getting the point.

This has been clarified at line 576-582. We hope the meaning is now clearer

Presumably it would be interesting to finish with a small recommendation what kind of stratospheric representation should be achieved in the future. How good should a QBO be? And how about the ability to reproduce the PDFs of warmings, etc. … Where do we want to be in a decade?

We now added some further pointers for improved stratospheric simulations in prediction systems as suggested by the reviewer. The manuscript now makes recommendations for: more research on initial stratospheric conditions (line 523-524), the need for a complete mechanistic understanding of strat-trop coupling (lines 536-537), the need to resolve tropical effects including the mysterious QBO-MJO link (lines 550-552), the need to implement simple ozone chemistry models in prediction systems (lines 567-568), and the need to resolve the signal-to-noise paradox (582-584).

In summary – the seamless nature of going from initial value dominated prediction to boundary value dominated (prediction and) projection should be far clearer. Even though I understand in the logic of

the paper that the last prediction chapter is called "multidecadal prediction" and needs to be far clearer that we really talk about projections that heavily reflect the chosen scenario. I do not doubt that even in projections the stratosphere has a crucial role – in particular for the "quality" of the teleconnections and their changes under climate change (given that the stratosphere cools when the troposphere heats up). Thus, I find lines 472-474 awkward and not well put into context. Here, also the role of composition beyond GHGs is certainly important (e.g. the recovery of the ozone hole in the southern hemisphere or changes in Sahara dust outbreaks in the northern hemisphere). Overall, the paper is a nice summary rationalising the importance of the stratosphere for weather and climate modelling (predictions and projections) that could do with some additional tuning before final publication following the comments above.

We have rephrased to use projections throughout and hopefully our new figure 1 clarifies the transition with lead time. We have also deleted lines 472-474.

**RC2:**

A review of recent developments in our understanding of the role of the stratosphere in surface weather and climate variability is presented, focusing on those aspects that provide improvements in long range prediction. A prime predictor arising from stratospheric variability in mid-latitudes appears to be the polar vortex strength, which itself is influenced by other teleconnection patterns (e.g., QBO, ENSO, MJO). Time scales considered are monthly, seasonal, annual, decadal, multi-decadal, which all exhibit co-variability patterns between the stratospheric polar vortex and the tropospheric jet.

Overall, this review reads well and the selection of included phenomena and relationships looks appropriate to me. It's great to see a combined effort from an extensive core list of researchers in the field.

Thank you for these encouraging comments.

That the stratosphere matters at the time and spatial scales considered in this review is, as far as I can tell, meanwhile well established. From that perspective the review seems overly strong about trying to convince the reader that the stratosphere does indeed matter. Arguing too much from the perspective that someone still needs to be convinced runs the risk of overstating the role of the stratosphere and overselling the point.

On the other hand this raises the question what the review is actually trying to achieve? It certainly represents a nice and comprehensive collection of those phenomena relevant for long range prediction that are influenced by the stratosphere. But in parts these aspects can already be found in previous reviews (e.g., Gerber et al., who review the importance of including a well-resolved stratosphere in weather and climate models; Kidston et al., who point out the similarity of S-T coupling on a range of time scales; Butler et al. (2019 book chapter) about the role of the stratosphere in sub-seasonal prediction). To me this calls for a better justification for the present review.

The present review is focused specifically on the role of the stratosphere in long range prediction as opposed to the more general reviews in Gerber et al and Kidston et al for example which focus more on the processes in stratosphere-troposphere coupling rather than the predictability questions addressed here. It was also solicited by the editors as a contribution to the associated 'Encyclopedia of Geosciences'.

I do think the material and expertise/experience on the author team offers a unique and welcome opportunity to synthesize our knowledge gained over the past ~20+ years in a way that offers new insights. For example, what can be said about the relative importance of the stratosphere compared to other sources of long range predictability? Does the stratosphere primarily act as an "integrator" of other sources of long range predictability (e.g., mid-lat weather regimes, MJO, ENSO, QBO) or does it have a more fundamental impact on predictability on its own (the latter perhaps more relevant for solar influences)? One way to synthesize our knowledge would be to create novel and meaningful schematics, e.g., to highlight the similarity across time scales and/or the interconnectedness of different climate system components. As it stands the review only includes 2 Figures: one on a recent SSW event (a case study) and one taken from another review article -- this seems a bit meager for an effort as this one.

We agree that more could be provided on these points and have now included an additional schematic figure on how the stratosphere contributes to long range prediction as you suggest. Please see the new Figure 1 which shows the contribution from different mechanisms involving the stratosphere and the transition from initial conditions to boundary conditions.

Another fundamental aspect is the distinction of limits of predictability due to sensitivity to initial conditions (e.g., on sub-seasonal time scales for weather or annual-to-decadal time scales for climate) vs. limits of long-term climate projections due to interactions across different components of the climate system (a very different animal). The presented material seems a bit superficial when it comes to this distinction and I'd strongly encourage the authors to revisit all related statements throughout the paper.

Thank you for this suggestion. We have now been clear about where the predictability arises mainly from initial conditions, where it arises due to boundary conditions and where it occurs through interaction with other parts of the climate system. See the new Figure 1, and in addition to the existing text on this point see also changes at lines 56-65, 112-113, 411-413, 426-427.

A few specific comments by line number:

line 36: "parallel advances" may hide the fact that the listed advances happened, at least in part, because of interactions across the involved communities

This is a fair point. However, our wording of the abstract does not preclude interaction so we added a comment at line 510-512 to emphasize the interaction between the communities.

line 45: whether the term "climate system" encompasses daily weather fluctuations is debatable; I'd suggest to avoid confusion here and start the sentence with "Daily weather fluctuations are thought to have ..."

Done

line 55: the way it's written it may sound as if SSWs are predictable; what is likely meant here is that the state of the stratosphere is (somewhat) predictable following a SSW -- please clarify

Rewritten as: "Some of the more prominent examples of stratospheric variability such as sudden stratospheric warmings and their subsequent impact on the stratosphere and the troposphere…"

line 114: suggest to change "affect" to "include"

Done

line 147ff: there's a more direct QBO-polar vortex connection, so it seems strange that the teleconnection via the MJO, which is much more indirect, gets mentioned first

This is simply because of timescale as the MJO is treated in the shorter (monthly section). This link is also now appearing to be more important than has been recognised until recently in the cited papers so we want to keep this emphasis.

line 157: "Other mechanisms" may sound confusing, because the preceding paragraphs were focused on tropospheric wave activity providing a source for stratospheric variability, whereas here you focus on the mechanisms around downward coupling.

This sentence was deleted.

Fig. 1: did you average over the 3 initialization dates; how did you compute the anomalies in panel b? please provide more detail about how these panels were produced

These are now described in more detail in the figure caption. The anomalies are calculated relative to hindcasts over the 1993-2016 period and yes, they are averaged over the initialisation dates.

Fig. 2: it would help to modify this schematic in such a way that the role of the stratosphere in the individual components shown stands out more clearly

The schematic is from another paper but please note the new Figure 1 which fills this requirement by showing only those components involving the stratosphere. We still retain the Figure as it shows all components (rather than just those involving the stratosphere) and therefore highlights land surface etc which was raised elsewhere in the reviews.

line 397: is "prediction" in this context still appropriate? or rather "projection"?

We have reworded this longer timescale to use 'projections' throughout, as also requested by the other reviewer.

line 505: "are responsible" sounds too strong to me -- something like "contribute to" seems more appropriate (the climate extremes can in principle happen with or without stratospheric influence)

We have said 'can lead to' to allow for similar extremes occurring for other reasons

line 518: "but occurred again" -- unclear what this refers to?

Apologies for the lack of clarity. This refers to the winter of 2020/21 so we have reworded this to "This is not generally reproduced in modelling systems (Garfinkel et al., 2012) but occurred in the recent La Nina winter of 2020/2021." and hope this is clearer.

line 528: ok, but perhaps important to mention that these nonlinear, state dependent impacts may be present for all biases in general, not just those involving the stratosphere

Done – this is no longer specific to the stratosphere

line 546: I agree that more research on the role of the stratosphere in the signal-to-noise paradox would be very useful; I think it'd be great for the general readership if the authors could expand on this idea a bit more

Thanks for this suggestion. We have added additional points and clarifications as suggested and the paragraph has been rewritten at lines 569-584.